# Validation of the Arabic Chronic Rhinosinusitis Patient-Reported Outcome (CRS-PRO): Translation and Cultural Adaptation

**DOI:** 10.3390/healthcare13030206

**Published:** 2025-01-21

**Authors:** Ameen Biadsee, Natalie Abu Amsha, Leigh J. Sowerby, Tomer Boldes, Firas Kassem

**Affiliations:** 1Department of Otolaryngology-Head and Neck Surgery, Meir Medical Center, Kfar Saba 4428164, Israel; aminbiadsee@tauex.tau.ac.il (A.B.); kassemfiras@gmail.com (F.K.); 2School of Medicine, Faculty of Medical and Health Sciences, Tel Aviv University, Tel Aviv 69978, Israel; 3Institute of Pathology, Samson Assuta-Ashdod University Hospital, Ashdod 7747629, Israel; natalie.a.amsha@gmail.com; 4Faculty of Medicine, Ben Gurion University of the Negev, Beersheba 8499000, Israel; 5Department of Otolaryngology-Head and Neck Surgery, Western University, London, ON N6A 3K7, Canada; leigh.sowerby@gmail.com

**Keywords:** chronic rhinosinusitis, patient-reported outcome measure, SNOT-22, CRS-PRO, quality of life

## Abstract

Background: The chronic rhinosinusitis patient-reported outcome (CRS-PRO) is a newly developed, disease-specific questionnaire designed for patients with CRS. This study focused on translating the CRS-PRO into Arabic, conducting cross-cultural adaptation and validation of the questionnaire, and assessing its reliability and validity. Methods: This prospective study involved 112 patients divided into CRS, functional endoscopic sinus surgery (FESS), and control groups. Participants completed the questionnaire at enrollment and again after one month. The Arabic version of the CRS-PRO was created following the International Society for Pharmacoeconomics and Outcomes Research guidelines for translation and cross-cultural adaptation. Results: This study included 74 males (66.1%) and 38 females (33.9%), with an average age of 37.4 ± 14.8 years. The Arabic CRS-PRO questionnaire has high internal consistency and reliability (Cronbach’s alpha 0.97). It also has strong discriminant validity in distinguishing between groups (ANOVA, *p* < 0.001). The assessment of test/retest symptom scores and their consistency over time confirmed the reliability of the CRS-PRO in differentiating CRS patients from healthy individuals and in monitoring surgical outcomes. This was validated through Pearson’s correlation coefficients (*p* < 0.01) and intraclass correlation (*p* < 0.0001). Conclusions: The Arabic version of the CRS-PRO proved simple, reliable, and valid. It showed high internal consistency, reliability, and strong discriminant validity in distinguishing between healthy individuals, CRS patients, and those pre- and post-FESS.

## 1. Introduction

Chronic rhinosinusitis (CRS) is a chronic, inflammatory condition affecting the sinonasal cavities, with a prevalence of 5–12% [1,2]. It is a multifactorial disease that significantly impacts patients’ general health-related quality of life (QoL) and profoundly affects functional well-being. CRS negatively influences sleep, mood, and work productivity [3,4]. Initial management involves intranasal corticosteroids and saline irrigation, with functional endoscopic sinus surgery (FESS) reserved for severe or refractory cases [4,5,6].

In the pediatric population, CRS often presents with a more heterogeneous etiology and less established endotyping [5,7]. QoL in this group is typically assessed using parent-reported questionnaires, such as the Sinus and Nasal Quality of Life Survey (SN-5) [8] and the Child Health Questionnaire-Parent Form 50 (CHQ-PF50) [9].

In adults, however, patient-reported outcome measures are essential tools for assessing the severity of disease, symptom burden, and patients’ subjective understanding of their illness [4,5,10,11]. According to a recent study, 61.4% of clinicians use symptom evaluation questionnaires for CRS, of which 67% are fellowship-trained specialists [12].

To assess CRS symptoms, medical or surgical treatments, and health-related QoL, the Rhinosinusitis Outcomes Measure-31 (RSOM-31) was introduced in 1995 [12]. To simplify its complexity, modified and abbreviated questionnaires such as the Sino-Nasal Outcome Test-20 (SNOT-20) and the 22-item Sino-Nasal Outcome Test (SNOT-22) were developed [5,13].

Although SNOT-22 has been widely adopted, its limitations include development before updated definitions of CRS, over-sensitivity to extra-nasal diseases, and lack of direct input from CRS patients [13,14,15]. The newly developed chronic rhinosinusitis patient-reported outcome (CRS-PRO) is a 12-item questionnaire designed in alignment with federal drug administration (FDA) guidelines for patient-reported outcome measures, enabling its use as an endpoint in future clinical trials. The CRS-PRO has been validated for CRS patients undergoing both medical treatment and FESS [14,15].

The cross-cultural adaptation, translation, and validation of questionnaires into different languages are critical for maintaining their validity and expanding global accessibility and impact. This study focused on translating the CRS-PRO questionnaire into Arabic, as well as validating and evaluating its reliability among Arabic-speaking CRS patients undergoing either medical treatment or FESS.

## 2. Methods

### 2.1. Ethical Considerations

This study was approved by the Meir Medical Center Institutional Review Board (IRB# MC-23-0082; 26/03/2023). All patients provided signed informed consent to participate.

### 2.2. Translation

Following the International Society for Pharmacoeconomics and Outcome Research (ISPOR) Task Force guidelines and principles for patient-reported outcome measures [16], the CRS-PRO questionnaire was translated and culturally adapted into Arabic. Two independent translators translated the original CRS-PRO questionnaire from English to Arabic. The resulting translations were then consolidated into one questionnaire and retranslated back into English by another independent translator. Finally, the retranslated version was reviewed by two independent reviewers fluent in both languages and verified to be congruent with the original intentions of the CRS-PRO questions.

### 2.3. Questionnaire

The CRS-PRO consists of 12 questions, each scored on a five-point scale from 0 to 4. The questions are organized into three parts: (1) physical symptoms, with 7 questions; (2) sensory impairment, with 1 question; and (3) psychosocial effects, with 4 questions.

### 2.4. Participants

This prospective study included adult patients (≥18 years) who were native Arabic speakers, capable of understanding and completing the questionnaire at baseline and again after one month. Participants were categorized into three groups:(1)CRS Group: Patients diagnosed with CRS, with or without nasal polyps, who met the criteria outlined in the European Position Paper on Rhinosinusitis and Nasal Polyps 2020 [5].(2)FESS Group: Patients also met CRS criteria and were scheduled for FESS. These individuals completed the questionnaire before surgery and repeated it one month after.(3)Control Group: Healthy individuals with no history of CRS, no prior nasal surgery, and no rhinologic complaints or need for intranasal medications, including intranasal corticosteroids [6].

Exclusion criteria included prior rhinologic surgery, sinonasal pathology other than CRS (with or without polyps), cognitive impairment (based on prior medical records), inability to complete the questionnaire, and pregnant or lactating women.

### 2.5. Statistical Analyses

All statistical analyses were performed using SPSS software, version 29.0 (IBM Corp., Armonk, NY, USA). Descriptive statistics were used to characterize the study participants, including frequencies and percentages for categorical variables, as well as means, standard deviations, and ranges for continuous and symptom variables. Univariate tests such as the Chi-square and one-way analysis of variance (ANOVA) were employed to examine differences in demographics and CRS-PRO scores among the three study groups: CRS, FESS, and control. Internal consistency, test/retest reliability, and validity of the CRS-PRO questionnaire were analyzed. Cronbach’s alpha coefficient was used to evaluate internal consistency.

Test/retest reliability was estimated by calculating the intraclass correlation coefficient (ICC) and Pearson’s correlation coefficient between test and retest instrument scores.

Concurrent validity and the performance of the instrument were assessed on the entire population and on the three subgroups: CRS, FESS, and control. Significance was determined at *p* < 0.05.

## 3. Results

Conducted between August 2023 and March 2024, this study included 112 participants, of which 74 were males (66.1%) and 38 were females (33.9%), with a mean age of 37.4 ± 14.8 years. The study cohort was divided into three groups: CRS (34 patients), FESS (30 patients), and control (48 patients). Table 1 summarizes the baseline demographics. The control group was younger than the CRS and FESS groups (*p* = 0.002).

## 4. Test/Retest Symptom Scores

The initial mean CRS-PRO scores were 2.8 ± 3.2 for the control group, 32.9 ± 7.8 for the FESS group, and 29.2 ± 10.7 for the CRS group (Table 2). All study participants completed the retest one month after the initial test. The retest scores were 1.2 ± 2.3 for the control group, 7.7 ± 3.3 for the FESS group, and 28.2 ± 13.2 for the CRS group.

The validity of the CRS-PRO questionnaire was evaluated by comparing the mean total scores among the three groups, which showed significant differences (*p* < 0.001, ANOVA).

Notably, CRS-PRO test scores for healthy participants were significantly lower in all three domains (‘physical symptoms’, ‘sensory impairment’, and ‘psychosocial effects’) compared to CRS patients, including the pre-surgery FESS group (*p* < 0.001, ANOVA). Additionally, retest scores for the CRS group were significantly higher than those for the healthy and post-operative FESS groups (*p* < 0.001, ANOVA). This pattern indicates that the CRS-PRO questionnaire has strong discriminant validity.

## 5. Reliability Assessment of the CRS-PRO

The internal consistency of the CRS-PRO questionnaire was confirmed by Cronbach’s coefficient alpha, which exceeded > 0.70 for all three groups and is considered adequate (Table 3). High internal consistency was observed in the CRS group, with Cronbach’s alpha values of 0.908 for the first test and 0.926 for the retest, and an overall value of 0.918. The overall Cronbach’s alpha for the total cohort was 0.97. Specifically, in the test/retest, Cronbach’s alpha values were 0.719 for the control group, 0.951 for the FESS group, and 0.918 for the CRS group, demonstrating good internal consistency of the CRS-PRO questionnaire.

The ICC and Pearson’s correlation coefficient were used to evaluate test/retest reliability for the total score and each domain. An ICC above 0.75 is considered good, and an ICC greater than 0.90 is regarded as excellent. For the CRS group, the ICC was 0.881 (*p* < 0.0001), demonstrating strong test/retest reliability. Pearson’s correlation also showed a significant correlation (*p* < 0.01) between initial and retest scores of the CRS-PRO (Table 3).

## 6. Discussion

This study evaluated the cross-cultural adaptation and validation of the Arabic version of the CRS-PRO. To date, only a few studies have translated and adapted this questionnaire from English into other languages [17,18]. All study participants completed the questionnaires without difficulty, indicating that the translation is an easy-to-administer self-reported questionnaire with good comprehension.

The results indicated high internal consistency and reliability of the Arabic version, with a Cronbach’s alpha of 0.97, surpassing the original version’s coefficient of 0.86 [14].

We observed strong discriminant validity for the CRS-PRO questionnaire in differentiating among groups (ANOVA, *p* < 0.001).

The test/retest reliability measures the questionnaire’s stability over time, demonstrating the ability to reproduce the same results if study participants retake the questionnaire. The reliability of the Arabic version is further supported by the analysis of the test/retest symptom scores and consistency over time. Our investigation indicated high test/retest reliability, confirmed by Pearson’s correlation coefficient (*p* < 0.01) and the ICC (*p* < 0.0001). The validation study showed that the Arabic version has good reliability between CRS patients and healthy individuals and in the responses provided after FESS.

We found that the Arabic version of the CRS-PRO is valid for CRS patients and post-FESS patients and is an effective instrument for measuring symptom changes after surgery [15].

Our results are consistent with those from the Chinese validation study, which reported excellent internal consistency and high reliability (Cronbach’s alphas of 0.936 and 0.81, respectively) and demonstrated good discriminant validity between CRS patients and healthy individuals [17].

## 7. Limitations

This study is subject to limitations. First, the sample size of each group was relatively small. Although statistical analyses revealed significant differences, the limited number of participants reduced the overall statistical power and may affect the generalizability of the results. Future research should include larger cohorts to confirm these initial findings and provide more robust evidence of the accuracy of the Arabic version of the CRS-PRO.

Second, there was a disparity in mean age among the groups: the control group, consisting of healthy individuals visiting for non-rhinologic concerns, was significantly younger than the CRS and FESS groups. This discrepancy likely resulted from convenience sampling, as younger adults seeking basic medical consultations were more readily enrolled. As a result, age was not perfectly balanced across groups, and the younger demographic in the control group could introduce potential bias. While statistical adjustments were performed, this difference may still affect the generalizability of the findings.

Despite these limitations, our data strongly support the validity and reliability of the Arabic version of the CRS-PRO in assessing symptom severity in patients with CRS.

## 8. Future Perspectives

Future research should investigate the correlation of the CRS-PRO with objective measures, such as endoscopic evaluations and imaging findings, to provide a more comprehensive assessment of disease severity and treatment outcomes. Longer-term follow-up studies are also recommended to evaluate the questionnaire’s reliability and responsiveness over time. Additionally, translating and validating the questionnaire in other languages and cultural contexts will enhance its global applicability and benefit diverse populations with CRS.

## 9. Conclusions

The CRS-PRO was successfully translated, validated, and adapted into Arabic. The Arabic version of the CRS-PRO demonstrated high internal consistency and reliability. Additionally, it effectively assessed and differentiated between healthy individuals, CRS patients, and those pre- and post-FESS, making it a valuable tool for evaluating patients before and after this surgery.

## Figures and Tables

**Table 1 healthcare-13-00206-t001:** Baseline demographics.

Variable	Control (*n* = 48)	FESS (*n* = 30)	CRS (*n* = 34)	Total (*n* = 112)	*p*-Value
Sex, *n* (%)					
Male	31 (64.6%)	22 (64.7%)	21 (70%)	74 (66.1%)	0.868
Female	17 (35.4%)	12 (35.3%)	9 (30%)	38 (33.9%)	
Age, years, mean (SD)	31.9 (9.1) ^a^	42.2 (17.5) ^b^	40.9 (16.2) ^b^	37.4 (14.8)	0.002
Age group, *n* (%)					
18–40	43 (89.6%) ^a^	19 (55.9%) ^b^	19 (63.3%) ^b^	81(72.3%)	0.002
41–90	5 (10.4%)	25 (44.1%)	11 (36.7%)	31 (27.7%)	

The percentages represent the column proportions. Differences in categorical variables (sex, age group) were tested using the Chi-square test. One-way ANOVA was used to compare continuous variables (age). Superscripts (^a.b^) indicate significant differences in post hoc tests, *p* < 0.05 (for Chi-square test, z-test with Bonferroni correction for comparisons of column proportions, and for one-way ANOVA, Tukey HSDs to examine average differences between all pairs).

**Table 2 healthcare-13-00206-t002:** Test/retest symptom scores.

Variable	Control (*n* = 48)	FESS (*n* = 30)	CRS (*n* = 34)	Total (*n* = 112)	*p*-Value
Initial test score					
Total CRS-PRO	2.8 (3.2)	32.9 (7.8)	29.2 (10.7)	18.9 (15.9)	<0.001 ^§^
Physical symptoms	2.0 (2.4)	19.2 (4.7)	16.8 (6.6)	11.1(9.2)	<0.001 ^§^
Sensing	0.1(0.5)	2.6 (1.5)	2.6 (1.3)	1.5 (1.6)	<0.001 ^§^
Psychologic	0.6 (1.2)	11.2 (3.5)	9.8 (4.2)	6.2 (5.8)	<0.001 ^§^
Retest score					
Total CRS-PRO	1.2 (2.3)	7.7 (3.3)	28.2 (13.2)	11.2 (13.8)	<0.001 ^†^
Physical symptoms	0.9 (1.7)	4.8 (2.0)	16.1 (6.4)	6.5 (7.6)	<0.001 ^†^
Sensing	0.0 (0.2)	0.7 (0.8)	2.5 (1.5)	0.9 (1.4)	<0.001 ^†^
Psychologic	0.3 (1.0)	2.2 (1.9)	8.9 (5.)	3.4 (4.7)	<0.001 ^†^

CRS, chronic rhinosinusitis; FESS, functional endoscopic sinus surgery. The total and subgroup scores are presented as the mean (standard deviation). ANOVA was used. ^§^ In post hoc tests, significant values are shown only for CRS vs. control and FESS vs. control. Comparisons between CRS and FESS were not significant. ^†^ In post hoc tests, all group comparisons resulted in a significant *p*-value.

**Table 3 healthcare-13-00206-t003:** Reliability assessment of the CRS-PRO.

Variable	Control (*n* = 48)	FESS (*n* = 30)	CRS (*n* = 34)	Total (*n* = 112)
Cronbach’s Alpha ^†^				
Test physical symptoms	0.635	0.687	0.852	0.942
Test psychologic	0.619	0.793	0.894	0.955
Total test	0.695	0.806	0.908	0.968
Retest physical symptoms	0.656	0.052	0.857	0.940
Retest psychologic	0.661	0.503	0.913	0.942
Total retest	0.717	0.402	0.926	0.967
All physical symptoms	0.661	0.911	0.854	0.944
All psychologic	0.638	0.921	0.905	0.952
All (test/retest)	0.719	0.951	0.918	0.970
Test/Retest Correlation (Pearson’s Correlation)				
Physical symptoms	0.609 ***	0.266	0.730 ***	0.662 ***
Sensing	0.330 *	0.167	0.753 ***	0.646 ***
Psychologic	0.216	−0.301	0.860 ***	0.618 ***
Total CRS-PRO	0.519 ***	−0.086	0.802 ***	0.657 ***
ICC				
Physical symptoms (F)	0.669 (3.667) ***	0.043 (1.485) ***	0.845 (6.407) ***	0.725 (4.079) ***
Sensing (F)	0.349 (1.548)	0.117 (1.330)	0.857 (6.916) ***	0.746 (4.527) ***
Psychologic (F)	0.348 (1.546)	−0.09 (0.592)	0.911 (12.284) ***	0.696 (4.080) ***
Total CRS-PRO (F)	0.604 (2.943) ***	−0.13 (0.883)	0.881 (8.315) ***	0.731 (4.727) ***

* *p* < 0.05 (2-tailed); *** *p* < 0.0001. ^†^ Subdomain ’Sensing’ (Q8) contains only 1 question; thus, the Cronbach’s alpha test was not applied.

## Data Availability

Data will be made available on request to the corresponding author.

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
