# Peer review of "Validation of the Arabic Chronic Rhinosinusitis Patient-Reported Outcome (CRS-PRO): Translation and Cultural Adaptation"

_healthcare, 2025, doi:10.3390/healthcare13030206_

Round 1
Reviewer 1 Report
Comments and Suggestions for Authors
Dear Editor
Thanks a lot for hard work. I read this article with interest. However, I have some concerns.
Kindly incorporate the responses within the manuscript to augment its overall quality.
The differences between chronic sinusitis in adults and children and the impact of chronic sinusitis on quality of life in adults and children should be stated in the introduction.
In the results, very small groups n=48,n=30n =34are analysed. a twofold increase in size is necessary to draw stastically significant conclusions in order to obtain adequate EBM power.
Literature should be supplemented with:https://doi.org/10.3390/jpm13040618; https://doi.org/10.3390/children8121133
Comments on the Quality of English LanguageThere are many spelling and language mistakes and the manuscript needs to be corrected by a native English speaker.
Author Response
We would like to extend our thanks for your thorough and thoughtful review of our manuscript. Your detailed observations and suggestions have greatly contributed to enhancing the clarity and quality of our work. We appreciate the time and effort you invested in this process and value your role in helping us refine our study. Comment 1: The differences between chronic sinusitis in adults and children and the impact of chronic sinusitis on quality of life in adults and children should be stated in the introduction. Response 1: Thank you for your insightful comment. We addressed your suggestion by including a discussion on the differences between chronic sinusitis in adults and children, as well as the impact of chronic sinusitis on quality of life in these populations. This addition includes the recommended references to strengthen the discussion. We are confident that this enhancement will provide valuable insights to the readers of Healthcare. The changes can be found on page 2, paragraph 2, lines 7-10. Comment 2: In the results, very small groups n=48,n=30n =34are analysed. a twofold increase in size is necessary to draw stastically significant conclusions in order to obtain adequate EBM power. Response 2: Thank you for your comment regarding the sample size. While we acknowledge that the groups were relatively small (n=48, n=30, n=34), the statistical analyses conducted demonstrated significant differences between the groups, providing valuable insights despite the limitations in sample size. We agree that larger studies are needed to confirm these findings and further validate our conclusions. We highlighted this in the Discussion section as a limitation and a direction for future research. The changes can be found on pages 10-11, paragraph 5, lines 24-29. Comment 3: There are many spelling and language mistakes and the manuscript needs to be corrected by a native English speaker. Response 3: The manuscript has been thoroughly reviewed and corrected by a native, English-speaking medical editor to ensure clarity, accuracy, and proper language use. We appreciate your attention to detail and have taken the necessary steps to address this concern.Reviewer 2 Report
Comments and Suggestions for Authors
Thank you for the opportunity of reviewing this manuscript entitled "
Validation of the Arabic Chronic Rhinosinusitis Patient-Reported Outcome: Translation and Cultural Adaptation"
This article is focused on the validation of a novel questionnaire to assess CRS burden on patients' quality of life. This is an appreciated topic in litterature
My suggestions are as follows:
- Description of inclusion criteria should be improved. "Control group individuals were selected from patients visiting the clinic for other issues such as CRS, rhinitis, or those requiring intranasal medications ". Controls should be chosen between healthy patient (as you state above). Rephrase this sentence. Also, have you defined cognitive impairment ? For what reason has pregnancy been considered exclusion criterion?
- Enhance the statistics description in your Methods chapter. You must disclose each statistical analysis you are going to perform for the evaluation of test-retest reliability, internal consistency, etc. Also, please consider that the current citation for SPSS "SPSS is IBM Corp., Armonk, NY, USA"
- Age difference is a limitation of your analysis. Please disclose this in your discussion
- The discussion doesn not provide sufficient insight into the background of your research. Include citations from articles which previously validated other questionnaires in different languages for CRS (doi: 10.3390/jpm12122010; doi: 10.1016/j.amjoto.2020.102775) Limitations are not disclosed. Future perspectives are not included.
Author Response
We would like to extend our thanks for your thorough and thoughtful review of our manuscript. Your detailed observations and suggestions have greatly contributed to enhancing the clarity and quality of our work. We appreciate the time and effort you invested in this process and value your role in helping us refine our study.
Comment 1: Description of inclusion criteria should be improved. "Control group individuals were selected from patients visiting the clinic for other issues such as CRS, rhinitis, or those requiring intranasal medications ". Controls should be chosen between healthy patient (as you state above). Rephrase this sentence. Also, have you defined cognitive impairment ? For what reason has pregnancy been considered exclusion criterion?
Response 1: Thank you for your valuable feedback. We appreciate your attention to the inclusion criteria and for highlighting the issue with the description of the control group and have revised this section to clarify it. The changes can be found in the Methods section on page 5, paragraph 4, and in the Results section on page 7, paragraph 1, lines 3-5. We also added a detailed explanation of cognitive impairment, noting that it was determined solely based on medical records, with no direct testing conducted by our team (page 6, paragraph 1, line 2. Regarding the exclusion of pregnant women, pregnancy was an exclusion criterion due to its known association with nasal congestion and pregnancy-related rhinitis. These conditions could potentially confound the results.
---
Comment 2: Enhance the statistics description in your Methods chapter. You must disclose each statistical analysis you are going to perform for the evaluation of test-retest reliability, internal consistency, etc. Also, please consider that the current citation for SPSS "SPSS is IBM Corp., Armonk, NY, USA"
Response 2: We enhanced the description of the statistical analyses in the Methods section, including detailed explanations of the tests performed to evaluate test-retest reliability, internal consistency, and other relevant statistical evaluations. Additionally, we updated the citation for SPSS to reflect the correct format: "SPSS version 29.0 (IBM Corp., Armonk, NY, USA." These revisions aim to improve the clarity and completeness of the Methods section. The changes can be found on page 6, paragraph2, lines 5-6.
---
Comment 3: Age difference is a limitation of your analysis. Please disclose this in your discussion.
Response 3: In the revised Discussion, we briefly addressed this as a notable limitation, emphasizing that the Control group, sampled predominantly from younger, healthy individuals, created an imbalance that may affect the broad applicability of our results. The changes can be found on pages 10-11, paragraph 5, lines 30-34.
---
Comment 4: The discussion doesn not provide sufficient insight into the background of your research. Include citations from articles which previously validated other questionnaires in different languages for CRS (doi: 10.3390/jpm12122010; doi: 10.1016/j.amjoto.2020.102775) Limitations are not disclosed. Future perspectives are not included.
Response 4: Thank you for your insightful comments. We incorporated the recommended references (doi: 10.3390/jpm12122010 and doi: 10.1016/j.amjoto.2020.102775) in the Introduction to enhance the background on patient-reported outcome measures, the changes can be found on page 4, paragraph 2, lines 9-11. In addition, we revised the Discussion to include a clearer statement of the study limitations and to outline future perspectives for research and clinical applications. The changes can be found on page 11, paragraph 2, lines 5-11.
Reviewer 3 Report
Comments and Suggestions for Authors
This manuscript was entitled as “Validation of the Arabic Chronic Rhinosinusitis Patient-Reported Outcome: Translation and Cultural Adaptation” This study aimed to explore the reliability and validity of the Arabic Chronic Rhinosinusitis Patient-Reported Outcome. The authors concluded that t The Arabic version of the CRS-PRO proved simple, reliable, and valid. It showed high internal consistency, reliability, and strong discriminant validity.
There is a major concern about this manuscript.
.
1. When the authors divided their participants into 3 groups: the controls, CRS patients, and those with FESS, the authors did not mention the calculation of the sample size for their participants.
Author Response
We would like to extend our thanks for your thorough and thoughtful review of our manuscript. Your detailed observations and suggestions have greatly contributed to enhancing the clarity and quality of our work. We appreciate the time and effort you invested in this process and value your role in helping us refine our study.
Comment 1: When the authors divided their participants into 3 groups: the controls, CRS patients, and those with FESS, the authors did not mention the calculation of the sample size for their participants.
Response 1: Thank you for highlighting this important aspect of our methodology. Empirical evidence suggests that a sample size of 30 or greater is generally adequate to ensure the applicability of the Central Limit Theorem (CLT); thereby, rendering the distribution of sample means approximately normal. We applied this principle when establishing the group sizes, aiming to maintain statistical validity and to enhance the reliability of our findings.
Reviewer 4 Report
Comments and Suggestions for Authors
The authors have evaluated a arabic translation of the CRS-PRO questionnaire. It is an important step for thee questionnaire to be widely used.
There are several issues that need to be sorted out
Was this the first version of the questionnaire. Then cognitive debriefing is very important to complete
Methods and Results - three different versions of controls have been mentioned. Please harmonise
Control group and the patients both had CRS. Please clarify.Who were the control subjects? What were the intranasal medications they needed?
In the paragraph above, it says with CRS. here it says without CRS?
Here in results, control group is referred to as healthy individuals. Then why did they come to the hospital?
Table 1: Please expand the demographic table to include more details such as clinical symptoms, any lab investigations, CT PNS details, severity of CRS for subjects in the groups were data is available, any co-morbidities such as asthma, diabetes, hypertension etc, smoking status so that the subjects are better characterized
Please recheck the results, as it is highly unlikely that the results of all physical symptoms of 0.661 and 0.638 in controls will lead to a final figure of 0.719
same for the other two groups; 0.911 and 0.921 can lead to 0.951
Almost all the values when totalled show similar results that the total is more than the individual components.
in total 0.968 is written as 0968
Author Response
We would like to extend our thanks for your thorough and thoughtful review of our manuscript. Your detailed observations and suggestions have greatly contributed to enhancing the clarity and quality of our work. We appreciate the time and effort you invested in this process and value your role in helping us refine our study.
Comment 1: Was this the first version of the questionnaire. Then cognitive debriefing is very important to complete
Response 1: Thank you for emphasizing the importance of cognitive debriefing. In our study, we followed the International Society for Pharmacoeconomics and Outcome Research (ISPOR) Task Force guidelines and principles for patient-reported outcome measures. Specifically, two independent translators initially translated the original CRS-PRO questionnaire from English to Arabic, and their versions were consolidated into one questionnaire. Subsequently, a third independent translator back-translated it into English. Finally, two independent, bilingual reviewers assessed the retranslated version to confirm its consistency with the original CRS-PRO. Our results indicate that the Arabic questionnaire was interpreted consistently, supporting its validity and reliability in this new context. Please see Methods, Translation Section
---
Comment 2: Methods and Results - three different versions of controls have been mentioned. Please harmonise.
Control group and the patients both had CRS. Please clarify. Who were the control subjects? What were the intranasal medications they needed?
In the paragraph above, it says with CRS. here it says without CRS?
Here in results, control group is referred to as healthy individuals. Then why did they come to the hospital?
Response 2: Thank you for highlighting the inconsistencies regarding the Control group. We apologize for any confusion. We clarified that intranasal medications, including intranasal corticosteroids (page 5, paragraph 4), were exclusively used by CRS patients as part of their standard treatment, whereas healthy individuals in the Control group did not receive any nasal therapy. This was corrected in the Methods section on page 5, paragraph 4, and in the Results section on page 7, paragraph 1, lines 3-5. We appreciate your thorough review and believe these revisions address the noted issue.
---
Comment 3: Table 1: Please expand the demographic table to include more details such as clinical symptoms, any lab investigations, CT PNS details, severity of CRS for subjects in the groups were data is available, any co-morbidities such as asthma, diabetes, hypertension etc, smoking status so that the subjects are better characterized
Response 3: As our primary objective was to validate the Arabic version of the CRS-PRO questionnaire, the data collection primarily focused on verifying its reliability rather than conducting a comprehensive clinical characterization of CRS. Since the CRS-PRO has already been established as an effective measure for CRS, our main goal was to confirm its validity and applicability in an Arabic-speaking population, not to extensively evaluate it as a diagnostic tool.
----
Comment 4: Please recheck the results, as it is highly unlikely that the results of all physical symptoms of 0.661 and 0.638 in controls will lead to a final figure of 0.719. same for the other two groups; 0.911 and 0.921 can lead to 0.951. Almost all the values when totalled show similar results that the total is more than the individual components.in total 0.968 is written as 0968
Response 4: Thank you for bringing this to our attention. We rechecked the data in consultation with our statistician, and we can confirm that the Cronbach’s alpha values reported are correct. As Cronbach’s alpha measures internal consistency, it can increase when more items (or domains) are combined, which explains why the total Cronbach’s alpha is higher than that of individual components. Additionally, we corrected the typographical error where “0.968” was inadvertently written as “0968.” We appreciate your careful review and have made the necessary corrections accordingly.
Round 2
Reviewer 1 Report
Comments and Suggestions for Authors
I accept in present form
Reviewer 2 Report
Comments and Suggestions for Authors
My inquiries have been adequately answered
Reviewer 3 Report
Comments and Suggestions for Authors
The authors have answered my concern. I do not have further comment.
Reviewer 4 Report
Comments and Suggestions for Authors
Authors have addressed the clarifications satisfactorily
No further comments